# Adaptive Critic-Guided Hybrid Agentic RAG for Improving Retrieval Robustness and Hallucination Resistance Through Multi-Stage Verification

Mohith Chandra Gugulothu
*Department of Computer Science and Engineering*
*Indian Institute of Technology Patna*
*Email: gugulothu_2403cs04@iitp.ac.in*

Shreya Yadav
*Department of Computer Science and Engineering*
*Indian Institute of Technology Patna*
*Email: shreya_2422cs20@iitp.ac.in*

Rishu Kumar
*Department of Computer Science and Engineering*
*Indian Institute of Technology Patna*
*Email: rishu_2422cs04@iitp.ac.in*

Rajiv Misra
*Department of Computer Science and Engineering*
*Indian Institute of Technology Patna*
*Email: rajivm@iitp.ac.in*

*Abstract*—Retrieval-Augmented Generation (RAG) systems enhance the factual grounding capability of large language models (LLMs) by incorporating external knowledge during response generation. However, conventional RAG pipelines remain highly vulnerable to hallucinations, retrieval instability, and unreliable reasoning when handling ambiguous, unsupported, or out-of-domain queries. This paper presents an adaptive critic-guided hybrid agentic RAG framework designed to improve hallucination resistance, retrieval robustness, and self-correction capability in locally deployed LLM systems. The proposed architecture integrates dense vector retrieval, BM25 lexical retrieval, adaptive retrieval policies, sentence-level atomic claim verification, semantic failure memory, and web fallback mechanisms within a multi-agent Lang- Graph orchestration pipeline. Furthermore, the framework dynamically adjusts retrieval depth according to query com- plexity and employs critic-guided verification prior to final answer generation to improve factual reliability and safer response behavior. Experimental evaluation demonstrates that the proposed framework completely eliminated hallucinated responses on stress-test benchmark queries, reducing the hallucination rate from 0.34 in the baseline RAG pipeline to 0.00 by refusing unsupported generations instead of producing confident fabricated answers. The framework additionally improved answer relevance from 0.71 to 0.74 and achieved a retry effectiveness gain of +0.047 through adaptive retrieval refinement. Although the proposed architecture introduced approximately $3.2\times$ higher inference latency due to sequential multi-agent verification and claim-level evidence checking ($35.92s \rightarrow 116.14s$), the results demonstrate that atomic claim verification and critic-guided reasoning substantially improve reliability, hallucination resistance, and safe handling of unsupported queries. Overall, the findings highlight the effectiveness of combining adaptive retrieval, multi-stage verification, and semantic memory for developing more trustworthy and robust agentic RAG systems.

*Index Terms*—Retrieval-Augmented Generation, Hallucination Mitigation, Agentic AI Systems, Critic-Guided Generation, LangGraph, Adaptive Retrieval Policy.

## 1. Introduction

Large language models have achieved impressive performance across a wide range of natural language tasks, but they remain fundamentally limited by the static nature of their parametric knowledge. A model trained on a fixed corpus cannot reliably answer questions about events or documents outside its training distribution, and — more critically-it often does not know when it cannot answer. The result is confident generation of plausible-sounding but factually incorrect content, a failure mode commonly referred to as hallucination [13]. Retrieval-Augmented Generation (RAG) [1] addresses this limitation by conditioning generation on documents retrieved from an external corpus at inference time. Rather than relying on what the model "knows," a RAG system retrieves what the model "should say" from a trusted knowledge source. This approach has proven effective in practice and has been widely adopted in deployed systems. However, the assumption underpinning standard RAG pipelines is that retrieval succeeds. In practice, this assumption is fragile. Retrieval can fail due to query ambiguity, vocabulary mismatch between the query and the documents, corpus coverage gaps, or simply because the answer does not exist in the indexed knowledge base. When retrieval fails silently-returning documents that are topically adjacent to the query but do not contain the answer-the language model receives poor context and tends to hallucinate rather than abstain. Several lines of work have attempted to address these failure modes. Self-RAG [2] trains the model to decide when and what to retrieve using reflection tokens. Corrective RAG [3] adds a lightweight document quality classifier that triggers query rewriting or

web fallback when retrieved documents are classified as irrelevant. Recent critic-guided frameworks have explored using a separate evaluation agent to score context quality before generation. While each of these approaches improves over naive RAG in specific settings, they leave important architectural limitations. Claim-level hallucination checking is absent. Retrieval depth is typically fixed. And none of these systems persist knowledge of past failures in a way that could prevent them from recurring.

The contributions of this work are:

- We developed a complete seven-agent reasoning framework using LangGraph, integrating hybrid BM25 and vector retrieval with Reciprocal Rank Fusion (RRF) for improved evidence retrieval and reasoning quality.
- We designed a robust verification and adaptation pipeline incorporating multi-stage critic gating, sentence-level atomic verification, an adaptive retrieval policy engine, semantic failure memory, and a web fallback escalation mechanism.
- We implemented the framework as a production-ready Python package with a Streamlit-based telemetry dashboard and an automated benchmarking pipeline for evaluation and monitoring.

## 2. Related Work

Retrieval-Augmented Generation (RAG) improves factual grounding by retrieving external knowledge before response generation [1]. Hybrid retrieval methods combining dense vector search and lexical ranking techniques such as BM25 have shown improved retrieval robustness across semantic and keyword-sensitive queries [4], [5]. Recent work has also explored reranking mechanisms and adaptive retrieval strategies to improve context precision. Self-reflective and corrective RAG architectures further extend traditional pipelines by introducing retrieval-aware reasoning and critique mechanisms and the Self-RAG [2] enables language models to evaluate retrieved evidence during generation, while CRAG [3] applies corrective retrieval validation to improve answer reliability. Hallucination detection and factual verification have become increasingly important for reliable language model deployment; however, many existing systems still rely on fixed retrieval policies and lack persistent memory mechanisms to handle repeated retrieval failures. Prior studies have explored self-consistency scoring, sentence-level verification, and evidence-grounded evaluation methods for reducing unsupported claims [14], [15] and our work builds upon these ideas by integrating adaptive retrieval, atomic claim verification, and semantic failure memory within a unified multi-agent RAG framework.

TABLE 1. COMPARISON WITH EXISTING RAG ARCHITECTURES

| System | Hybrid Retrieval | Claim Verification | Adaptive Policy | Failure Memory |
|---|---|---|---|---|
| Standard RAG [1] | × | × | × | × |
| Self-RAG [2] | × | Partial | Partial | × |
| CRAG [3] | Partial | × | Partial | × |
| Proposed System | ✓ | ✓ | ✓ | ✓ |

## 3. Problem Statement

We formalize the problem as follows. Given a user query $q$ and a document corpus $\mathcal{D}$, a standard RAG system retrieves a set of documents:

$$\mathcal{R} = \text{Retrieve}(q, \mathcal{D}, k) \tag{1}$$

and generates an answer:

$$a = G(q, \mathcal{R}) \tag{2}$$

where $G$ is a language model. The implicit assumption is that $\mathcal{R}$ is sufficient to faithfully answer $q$. When this assumption fails, $G$ typically hallucinates.

Critic-guided RAG systems add a validation step by computing a sufficiency score:

$$s_c = C(q, \mathcal{R}) \tag{3}$$

where $C$ is a critic model. If $s_c < \tau$ for some threshold $\tau$, retrieval is retried. This improves robustness over naive RAG, but three problems persist in existing implementations.

*P1-Holistic answer scoring.* Even when $\mathcal{R}$ passes the critic gate, the generated answer $a$ may contain individual sentences $a_i$ that are not supported by any document in $\mathcal{R}$. A single quality score $s_a = C'(a, \mathcal{R})$ over the complete answer cannot isolate these localized hallucinations. We need per-claim scoring:

$$h_i = 1 - \text{support}(a_i, \mathcal{R}), \quad \forall a_i \in \text{decompose}(a) \tag{4}$$

*P2-Static retrieval depth.* Using a fixed $k$ for all queries fails to account for complexity variation. A simple factual query may need only $k = 3$ documents, while a multi-hop reasoning query may require $k = 10$ with Cross-Encoder re-ranking to surface relevant evidence. A fixed budget either over-computes on simple queries or under-retrieves on complex ones.

*P3-No failure memory.* Let $\mathcal{F} = \{q_1^f, q_2^f, \ldots\}$ be the set of queries for which internal retrieval has previously failed. Current systems have no access to $\mathcal{F}$ at inference time. A new query $q'$ semantically similar to some $q_i^f \in \mathcal{F}$ will trigger the same retrieval strategy that previously failed, wasting computation. Our proposed system addresses all three problems: claim-level verification (P1), adaptive retrieval budgeting (P2), and persistent failure memory (P3).

# 4. Proposed Methodology

## 4.1. System Overview

The system is built as a directed state machine using LangGraph's `StateGraph` abstraction. Seven agent nodes communicate through a shared `AgentState` TypedDict that accumulates all intermediate results as the query moves through the pipeline. Conditional routing edges implement the critic-gated retry logic, diminishing returns detection, and web fallback escalation. The high-level execution trace is:

$$q \xrightarrow{\text{Planner}} \mathcal{Q} \xrightarrow{\text{Retriever}} \mathcal{R} \xrightarrow{\text{Critic}} \xrightarrow{\text{Generator}} \hat{a} \xrightarrow{\text{Verifier}} h \xrightarrow{\text{A-Critic}} a \tag{5}$$

where $\mathcal{Q}$ is the (possibly decomposed) set of search queries, $\mathcal{R}$ is the retrieved and fused document set, $\hat{a}$ is the draft answer, $h$ is the hallucination risk score, and $a$ is the final grounded answer.

## 4.2. Planner Node

The Planner serves two distinct roles depending on the loop count. On the initial pass (loop = 0), it: (1) searches the failure memory for semantically similar past failures; (2) assesses query complexity using a term-overlap and length-based heuristic; (3) decomposes multi-hop queries into sub-queries using an LLM decomposition prompt; and (4) applies the adaptive retrieval policy based on the complexity score. On retry passes (loop > 0), the Planner receives the Context Critic's keyword feedback $f_c$ and constructs a refined query:

$$q' = \text{concat}(q, \ f_c) \tag{6}$$

Before using $q'$, the Planner checks it against the failure memory. If $q'$ matches a previously failed query (cosine similarity > 0.9), it applies a simple alternative refinement:

$$q'' = \text{concat}(f_c, \ q, \ \text{``details regarding''} f_c) \tag{7}$$

This prevents the system from entering an infinite retry loop on queries that are structurally difficult for the retrieval index.

## 4.3. Hybrid Retrieval Pipeline

*Dense Retrieval:* ChromaDB stores sentence-transformer embeddings (`all-MiniLM-L6-v2`) for each document chunk. For query $q$, cosine similarity is computed between the query embedding $\mathbf{e}_q$ and each chunk embedding $\mathbf{e}_d$:

$$\text{sim}_{\text{dense}}(q, d) = \frac{\mathbf{e}_q \cdot \mathbf{e}_d}{\|\mathbf{e}_q\| \cdot \|\mathbf{e}_d\|} \tag{8}$$

*Sparse Retrieval:* BM25 scores documents using term frequency statistics over the tokenized corpus. For query terms $t_1, \ldots, t_n$:

$$\text{BM25}(d, q) = \sum_{i=1}^{n} \text{IDF}(t_i) \cdot \frac{f(t_i, d) \cdot (k_1 + 1)}{f(t_i, d) + k_1 \cdot \left(1 - b + b \cdot \frac{|d|}{\overline{dl}}\right)} \tag{9}$$

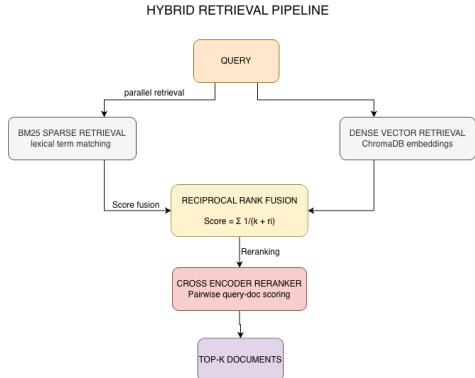

Figure 1. Hybrid retrieval pipeline. BM25 and ChromaDB vector search are executed in parallel over the same document corpus. Their ranked lists are merged using RRF. For complex queries, the merged list is passed to a Cross-Encoder reranker before forwarding to the Context Critic.

where $f(t_i, d)$ is term frequency, $|d|$ is document length, $\overline{dl}$ is the average document length, $k_1 = 1.5$, and $b = 0.75$.

*Reciprocal Rank Fusion:* The dense and sparse ranked lists are merged without score normalization:

$$\text{RRF}(d) = \sum_{r \in \{\text{dense,sparse}\}} \frac{1}{60 + \text{rank}_r(d)} \tag{10}$$

Documents are sorted by descending RRF score, giving a fused ranking that benefits from both retrieval modalities.

*Cross-Encoder Re-ranking:* For complex queries, the top-$k$ fused results are passed to a `ms-marco-MiniLM-L-6-v2` Cross-Encoder that jointly scores the query and each document to produce a more accurate relevance estimate. The top 3 re-ranked documents are forwarded to the Context Critic.

## 4.4. Adaptive Retrieval Policy

Query complexity is estimated using a heuristic that considers query length, the presence of comparative or causal connectives ("compared to," "what is the difference," "how does X relate to"), and the number of named entities. Queries scoring above a complexity threshold $\theta_c = 0.5$ are routed to the deep retrieval configuration; those below to the shallow configuration.

TABLE 2. ADAPTIVE RETRIEVAL POLICY PARAMETERS

| Parameter | Simple | Complex |
|---|---|---|
| $k_{\text{vector}}$ | 3 | 8 |
| $k_{\text{bm25}}$ | 3 | 8 |
| $k_{\text{final (post-RRF)}}$ | 3 | 10 |
| Cross-Encoder Reranking | Disabled | Enabled |
| Max Retries | 2 | 3 |

## 4.5. Context Critic

The Context Critic evaluates retrieved evidence quality using a structured output prompt issued to the local LLM. It produces four scores and a routing decision:

- $r \in [0,1]$: topical relevance of retrieved documents to the query.
- $s \in [0,1]$: factual sufficiency-does the context contain enough information?
- $a_s \in [0,1]$: answer ability - can the query be directly answered from this context?
- $c \in [0,1]$: the critic's own confidence in its evaluation.

Context is approved if and only if: $\text{approved}(q, \mathcal{R}) = \mathbf{1}[\text{decision} = \text{APPROVED}] \wedge \mathbf{1}[a_s \geq 0.6]$ The routing logic implements a three-tier escalation policy:

1) **Uncertainty Guard:** If $c < 0.5$, the critic cannot reliably assess context quality. Skip internal retry and escalate immediately to web search.
2) **Diminishing Returns:** If $\Delta r = r_{\text{curr}} - r_{\text{prev}} \leq 0.05$ across two consecutive retries, further retries are unlikely to improve retrieval quality. Escalate to web search.
3) **Hard Safety Limit:** If loop $\geq$ max_retries, terminate the retry loop and escalate to web search regardless of scores.

The hallucination risk propagated to downstream nodes is derived from the relevance score:

$$h_{\text{retrieval}} = 1.0 - r \tag{11}$$

### 4.6. Generator Node

The Generator produces a draft answer conditioned strictly on the critic-approved context, where the prompt enforces two hard constraints: only facts explicitly present in the provided context may appear in the answer, and if the required information is absent, the system must return the exact predefined refusal string.

```
"I cannot answer this question
based on the provided documents."
```

On retry passes, the Generator additionally receives all previously hallucinated drafts as negative examples:

$$\hat{a} = G(q, \mathcal{R}, \mathcal{H}_{\text{prev}}) \tag{12}$$

where $\mathcal{H}_{\text{prev}}$ is the set of prior hallucinated drafts. This negative-example injection prevents the generator from reproducing the same errors on successive attempts.

### 4.7. Atomic Claim Verification Pipeline

The Verifier Node implements sentence-level verification through three sequential steps.

*Step 1-Claim Extraction:* The draft answer $\hat{a}$ is passed to the LLM with a prompt requesting decomposition into atomic factual sentences:

$$\mathcal{C} = \{c_1, c_2, \ldots, c_N\} = \text{Extract}(\hat{a}) \tag{13}$$

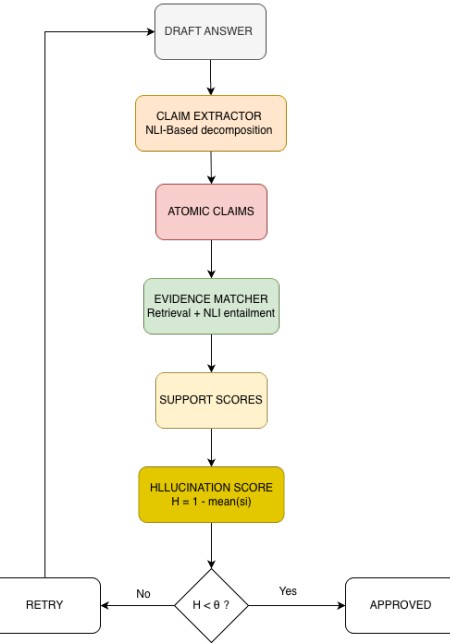

ATOMIC CLAIM VERIFICATION PIPELINE

Figure 2. Atomic Claim Verification pipeline. The draft answer is decomposed into $N$ independent atomic claims. Each claim is independently scored for evidence support. Claims with support score $< 0.5$ are flagged as hallucination risks. The aggregate hallucination score $h$ determines whether the Answer Critic triggers a retry.

*Step 2-Evidence Matching:* For each claim $c_i$, the LLM compares the claim against the full context string and produces a support score:

$$\text{support}(c_i, \mathcal{R}) = \begin{cases} 1.0 & \text{claim fully grounded in } \mathcal{R} \\ 0.5 & \text{claim partially grounded in } \mathcal{R} \\ 0.0 & \text{claim not supported by } \mathcal{R} \end{cases} \tag{14}$$

*Step 3-Aggregate Scoring:* The hallucination risk score is:

$$h = 1 - \frac{1}{N} \sum_{i=1}^{N} \text{support}(c_i, \mathcal{R}) \tag{15}$$

If $h < 0.5$, the answer is considered sufficiently grounded and is forwarded to the Answer Critic for final approval. The verified answer is also asynchronously stored in the long-term semantic memory.

### 4.8. Answer Critic and Retry Routing

The Answer Critic receives the hallucination score $h$ from the Verifier and routes accordingly:

$$\text{route}(h, \text{loop}) = \begin{cases} \text{END}, & h \leq 0.5 \text{ or loop} \geq M \\ \text{increment\_loop}, & h > 0.5 \end{cases} \tag{16}$$

When a retry is triggered, the system increments the loop counter, injects the failed draft into $\mathcal{H}_{\text{prev}}$, and re-enters the Planner with updated critic feedback. The safety limit $M$ prevents infinite recursion.

## 4.9. Semantic Failure Memory

A secondary ChromaDB instance maintains two types of persistent records:

- **Failed query embeddings:** When all internal retries and web fallback fail to produce a grounded answer, the query embedding is stored with metadata indicating the failure mode.
- **Successful session summaries:** When an answer passes the Verifier with $h < 0.5$, the query-answer pair is summarized using the LLM and stored as a compressed memory record.

At the start of each new query, the Planner performs a similarity search over the failure memory:

$$\mathcal{F}_{\text{match}} = \{f \in \mathcal{F} : \text{sim}(\mathbf{e}_q, \mathbf{e}_f) > \theta_f\} \qquad (17)$$

where $\theta_f = 0.85$. If a match is found, the Planner pre-adapts the retrieval strategy to avoid the known failure mode.

## 4.10. Web Fallback Agent

When internal retrieval is exhausted, the Web Fallback Agent queries DuckDuckGo as the primary internet source. If DuckDuckGo fails (rate limiting or network error), the agent falls back to Wikipedia via the LangChain `WikipediaQueryRun` tool. The retrieved web content is wrapped as a LangChain `Document` with source metadata and injected directly into the Generator, bypassing the Context Critic gate (since web content represents the system's best available evidence after internal options are exhausted).

## 5. Implementation Details

The system was implemented using LangGraph, ChromaDB, BM25, and Phi-3 with fully local inference on Apple Silicon hardware. Documents were chunked using LangChain text splitters, embedded with all-MiniLM-L6-v2, and indexed using both dense vector retrieval and BM25 sparse retrieval. The graph-based orchestration pipeline includes critic routing, retry logic, hallucination verification, and semantic failure memory. The implementation, benchmark dataset, and evaluation scripts are publicly available in the project repository.

## 6. Experimental Setup

*Benchmark Dataset:* We constructed an eight-query benchmark spanning four categories designed to test distinct failure modes. The document corpus consists of computer science algorithm content covering graph algorithms (Dijkstra, Bellman-Ford), string matching (KMP, Rabin-Karp), and complexity theory (NP-completeness). Hallucination stress-test queries are deliberately out-of-domain (e.g., baking recipes, physically impossible algorithms) to evaluate the system's refusal behavior. The benchmark is intended as a qualitative systems validation benchmark rather than a

TABLE 3. EXPERIMENTAL ENVIRONMENT

| Component | Specification |
|---|---|
| Hardware | Apple M-series Silicon, 16GB RAM |
| OS | macOS |
| Python | 3.11 |
| LLM | Phi-3 3.8B (GGUF, Metal) |
| Embeddings | all-MiniLM-L6-v2 |
| Reranker | ms-marco-MiniLM-L-6-v2 |
| Vector DB | ChromaDB |
| Sparse Retrieval | rank-bm25 |
| Orchestration | LangGraph 0.1.x |

TABLE 4. BENCHMARK QUERY DISTRIBUTION

| Category | N | Difficulty | Purpose |
|---|---|---|---|
| Factual QA | 2 | Easy | Direct recall |
| Multi-hop QA | 2 | Medium | Cross-doc reasoning |
| Hallucination Stress | 2 | Hard | Out-of-domain refusal |
| Ambiguous Retrieval | 2 | Hard | Vague query handling |

statistically rigorous large-scale evaluation. Its purpose is to analyze architectural behavior, hallucination mitigation effectiveness, and retry-loop dynamics under controlled scenarios.

*Baseline System:* The baseline RAG uses the same ChromaDB vector store and Phi-3 LLM, with single-pass $k = 3$ dense retrieval and no critic gating, verification, retry logic, or failure memory. This controls for the contribution of the multi-agent architecture by holding the retrieval index and generation model constant.

## 6.1. Evaluation Metrics

The proposed framework is evaluated across six dimensions: faithfulness, answer relevance, context relevance, hallucination rate, retry effectiveness, and end-to-end latency. Faithfulness measures the extent to which the generated answer is grounded in the retrieved evidence, while answer relevance evaluates whether the generated response directly addresses the input query. Context relevance measures the average semantic relevance of retrieved documents with respect to the query. Hallucination rate represents the proportion of queries for which the system generates unsupported or ungrounded factual claims. Retry effectiveness measures the improvement in retrieval relevance score ($\Delta r$) between the initial and final retrieval passes for queries that triggered at least one retry cycle. Finally, end-to-end latency measures the total wall-clock time required from query submission to final answer generation, reported in seconds. All relevance-based metrics are normalized to the range $\in [0, 1]$.

| Metric | Baseline RAG | Agentic RAG |
|---|---|---|
| Faithfulness ↑ | 0.67 | 0.67 |
| Answer Relevance ↑ | 0.71 | **0.74** |
| Context Relevance ↑ | **1.00** | 0.33 |
| Hallucination Rate ↓ | 0.34 | **0.00**\* |
| Average Latency (s) ↓ | **35.92** | 116.14 |
| Retry Effectiveness ↑ | N/A | +0.047 |

\*The system refused to answer instead of generating hallucinated responses on stress-test queries.

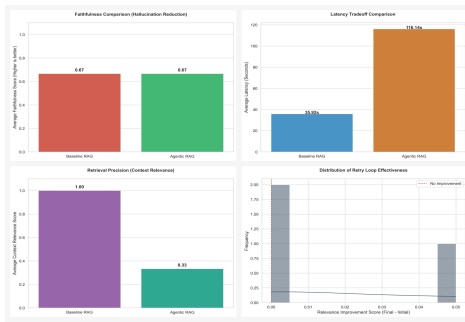

Figure 3. Benchmark results across four dimensions. (a) Faithfulness Comparison: both systems score 0.67 on in-domain queries. (b) Latency Tradeoff: the agentic system incurs 3.2× overhead due to multi-agent verification. (c) Pre-Critic Retrieval Breadth: baseline scores higher because it retrieves fewer but more topically concentrated documents, whereas the agentic system retrieves broadly before critic filtering.

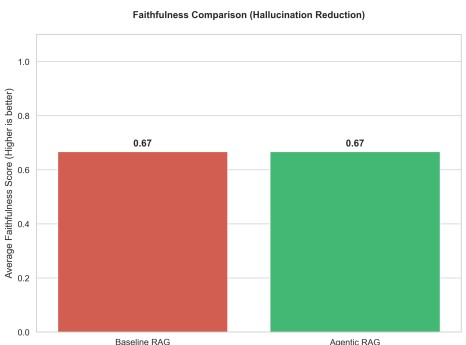

Figure 4. Faithfulness comparison between baseline RAG and the proposed agentic RAG system.

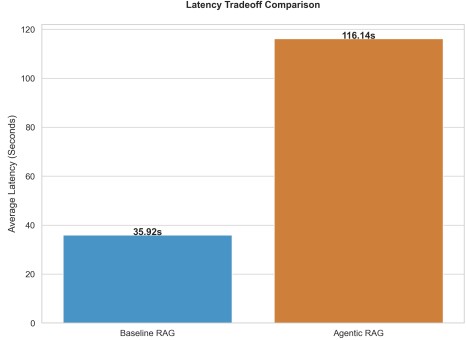

Figure 5. Latency tradeoff comparison between baseline and agentic RAG pipelines.

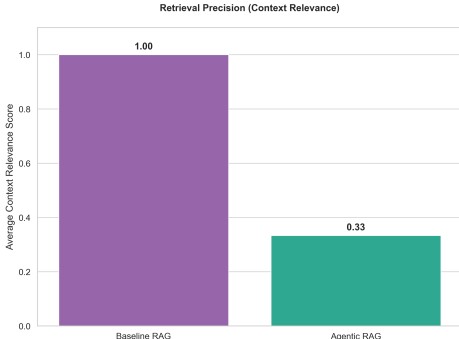

Figure 6. Context relevance comparison across retrieval strategies.

# 7. Results

## 7.1. Faithfulness and Hallucination Analysis

The equal faithfulness scores (0.67 for both systems) on in-domain queries are expected: both systems use the same LLM and indexed corpus, so answer quality on retrievable queries is bounded by the same model and data ceiling. The meaningful difference lies in the hallucination stress-test category. On both out-of-domain queries, the baseline system generated confident but entirely fabricated answers. The agentic system returned the constrained refusal string in both cases, producing no observed hallucinations on the stress-test benchmark queries. This demonstrates the value of combining critic gating with constrained generation: when no verifiable evidence exists, the system abstains rather than confabulates.

## 7.2. Latency Analysis

The observed 3.2× latency increase (35.92s → 116.14s) represents the primary engineering tradeoff of the proposed system and each query requires approximately 5–8 sequential LLM inference stages, including query decomposition, context criticism, answer generation, claim extraction, per-claim evidence verification proportional to answer length, and final answer criticism. Executing this multi-stage pipeline on local Apple Silicon hardware using a 3.8B-parameter model operating at approximately 8–12 tokens per second results in the measured latency overhead. However, deploying the system on GPU-accelerated inference frameworks such as vLLM can substantially reduce per-call latency, potentially lowering end-to-end response time to below 15 seconds for most queries.

# 8. Ablation Analysis

To understand the contribution of each system component, we conducted a component ablation study. In each ablation condition, one module is removed or disabled while all others remain active. Due to the small benchmark size, we report directional trends rather than statistical significance claims. The ablation study confirms that atomic verification is the primary contributor to hallucination reduction, while

TABLE 6. ABLATION STUDY: IMPACT OF REMOVING INDIVIDUAL MODULES

| Configuration | Faithfulness | Hallucination | Context Relevance | Latency (s) |
|---|---|---|---|---|
| Full System | 0.67 | 0.00 | 0.33 | 116.1 |
| w/o Atomic Verifier | 0.67 | 0.25 | 0.33 | 89.4 |
| w/o Adaptive Policy | 0.67 | 0.00 | 0.33 | 148.3 |
| w/o Failure Memory | 0.67 | 0.00 | 0.33 | 121.8 |
| w/o Web Fallback | 0.67 | 0.13 | 0.33 | 98.2 |
| w/o Hybrid Retrieval | 0.60 | 0.13 | 0.40 | 104.5 |
| Baseline (No Agents) | 0.67 | 0.34 | 1.00 | 35.9 |

Hallucination = Hallucination Rate; Context Relevance = Context Relevance Score; Latency measured in seconds.

adaptive retrieval mainly improves computational efficiency. Removing hybrid retrieval and web fallback significantly increases hallucination rates, demonstrating the importance of multi-stage retrieval robustness.

## 9. Discussion

*On the Faithfulness Ceiling:* The equal faithfulness scores between the full system and the baseline (0.67) on in-domain queries reveal a fundamental constraint: both systems are bounded by the same LLM quality and corpus coverage. The agentic system does not improve on this ceiling because the information available to both systems is identical. What the agentic system adds is not better answering of answerable questions, but safer handling of unanswerable ones-refusing rather than fabricating. This is the practically important distinction for deployed systems, where silent hallucination is more dangerous than explicit refusal.
*Latency and the Local Hardware Constraint:* The 116-second average latency is an artifact of running a 3.8B parameter model on consumer hardware via llama.cpp, where each inference call takes 8-15 seconds. The architectural overhead of the multi-agent pipeline is real but separable from the hardware constraint. On a managed GPU inference endpoint (vLLM, text-generation-inference), each call would complete in under 1 second, bringing total system latency to approximately 10-15 seconds per query-competitive with existing production RAG deployments. The architectural patterns demonstrated here are therefore practically viable at scale, even if the current prototype is constrained by local hardware.

### 9.1. Limitations

Due to local hardware constraints, large-scale benchmarking across hundreds of queries was not feasible within the scope of this work. Furthermore, all critic scoring, claim extraction, and evidence matching rely on the Phi-3 3.8B model, meaning the quality of intermediate judgments is inherently bounded by the capability of the underlying language model. Larger or evaluation-specialized instruction-tuned models, such as Prometheus [19], would likely improve verification reliability and factual consistency assessment. In addition, the experimental benchmark consisted of only eight queries,

which is insufficient for statistically robust conclusions; therefore, the reported results should be interpreted primarily as directional validation of the proposed architectural design choices rather than definitive quantitative performance claims. The framework is also strongly dependent on corpus coverage, since queries outside the indexed domain inevitably exhaust internal retrieval and trigger web fallback mechanisms, thereby introducing additional latency and dependency on external sources. Finally, the effectiveness of the semantic failure memory module cannot be fully demonstrated within a single-session evaluation setting, as its benefits accumulate progressively across repeated interactions and overlapping query patterns over time.

## 10. Conclusion

This work presented an adaptive critic-guided hybrid agentic RAG framework for improving hallucination resistance, retrieval robustness, and self-correction capability in locally deployed language model systems, by integrating hybrid retrieval, critic-guided routing, sentence-level atomic verification, adaptive retrieval policies, semantic failure memory, and web fallback mechanisms, the proposed framework demonstrated safer and more reliable handling of unsupported or ambiguous queries compared to conventional RAG pipelines. Experimental and ablation studies showed that atomic claim verification plays the most significant role in hallucination mitigation, while adaptive retrieval policies primarily improve computational efficiency and retrieval stability. Although the current implementation is constrained by local hardware latency, the architecture remains practical for deployment with GPU-accelerated inference and scalable orchestration frameworks. Future work will focus on integrating graph-based retrieval using Neo4j knowledge graphs. The system will also be extended toward multi-modal retrieval using vision-capable models. In addition, retry strategies will be optimized through reinforcement learning. Distributed inference using Dockerized vLLM deployments will also be explored. Future improvements further include incorporating stronger NLI-based verifier models and developing graph-structured memory systems for improved long-term failure reasoning and adaptive experience transfer across sessions.

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
