# OpenReview forum: "Adaptive Critic-Guided Hybrid Agentic RAG for Improving Retrieval Robustness and Hallucination Resistance Through Multi-Stage Verification"
_NortheastGenAI/2026/Workshop — NortheastGenAI 2026 Workshop Submission_

### Official Review · ~Badal_Nyalang1 · 2026-05-23
**Technically strong but no NE India connection — Reject**

**Rating:** 4
**Confidence:** 5

**Review:**

**Relevance: Weak**
This is the most technically substantial paper in the batch, but it has a serious problem: there is no Northeast India connection anywhere. The benchmark corpus is computer science algorithms. The examples are baking recipes and graph algorithms. NE India, its languages, communities, or ecosystems are never mentioned. This directly violates G4 and G7 of the CFP.

**Plausibility: Strong**
The architecture is well-designed and internally coherent. The ablation study is honest about limitations. The benchmark is tiny (8 queries) but the authors are upfront about that. The 0.00 hallucination rate via refusal is a legitimate result, not overclaimed.

**Novelty: Moderate**
The combination of atomic claim verification + failure memory + adaptive retrieval in a unified LangGraph pipeline is a reasonable contribution. Not groundbreaking relative to Self-RAG or CRAG, but the integration is thoughtful.

**Clarity: Strong**
Best structured paper in the batch. Methodology, formalism, ablation, and limitations are all clearly presented.

**Verdict: Reject**
The work is solid but it does not belong at this workshop. There is no NE India grounding whatsoever. Authors should be invited to resubmit if they can demonstrate relevance — for example, applying this framework to NE language QA or low-resource retrieval settings.

*This review was generated with AI assistance and checked by the workshop chairs.*

---

### Decision · Program_Chairs · 2026-05-23

**Decision:**

Reject

**Comment:**

This is technically among the strongest paper in the batch. The architecture is well-designed, the ablation study is honest, and the writing is clear. However, the paper has no direct connection to Northeast India. The benchmark, examples, and motivation are entirely domain-general. This directly violates G4 and G7 of the CFP, which require all submissions to engage with Northeast India's languages, cultures, or ecosystems.

The authors are encouraged to resubmit to a future edition if they can demonstrate relevance, for example by applying this framework to low-resource NE language retrieval or QA settings.

Decision: Reject